# A lanthanide MOF with nanostructured node disorder

Sarah L. Griffin[1,6], Emily G. Meekel [2,6], Johnathan M. Bulled[3,6], Stefano Canossa [4], Alexander Wahrhaftig-Lewis[1], Ella M. Schmidt [5] & Neil R. Champness [1]✉

Structural disorder can be used to tune the properties of functional materials and is an important tool that can be employed for the development of complex framework materials, such as metal-organic frameworks. Here we show the synthesis and structural characterization of a metal-organic framework, UoB-100(Dy). Average structure refinements indicate that the node is disordered between two orientations of the nonanuclear secondary building unit (SBU). By performing 3D diffuse scattering (DS) analysis and Monte Carlo (MC) simulations, we confirm the presence of strong correlations between the metal clusters of UoB-100(Dy). These nodes assemble into a complex nanodomain structure. Quantum mechanical calculations identify linker strain as the driving force behind the nanodomain structure. The implications of such a nanodomain structure for the magnetic, gas storage, and mechanical properties of lanthanide MOFs are discussed.

Over the past century, structural disorder has been established as a useful tool for the tuning of physical properties of functional materials. Disorder in $La_{1-x}Ca_xMnO_3$ and $Cu_xMn_{1-x}$, for example, is known to result in colossal magnetoresistance (CMR)[1–3] and spin-glass behaviour[4,5], respectively. Although such structural disorder can be random, it is often found to be correlated[6,7]. Depending on the interactions which drive these correlations, 'nanodomains' *i.e.* regions of local order, may emerge[8–17]. In the relaxor ferroelectric $PbMg_{1/3}Nb_{2/3}O_3$ (PMN), local polar regions of a few nanometres wide are responsible for its ferro- and piezo-electricity[8–14]. The transport properties of perovskite solar-cell materials are similarly linked to such structural nanodomains[17]. Explaining the origin of these behaviours is challenging, as the structures of these disordered materials span multiple length-scales[18]. To study nanostructures crystallographically, one must look at the weak, diffuse scattering (DS) signal running between and on top of the Bragg-peaks, as it reports on the correlations between disordered degrees of freedom.

The presence and significance of disorder in metal-organic frameworks (MOFs) have only recently gained widespread awareness[19]. Disorder in MOFs is not uncommon, however, as many MOFs will exhibit linker- or cluster-defects to some extent. This vacancy disorder tends to be correlated, as one defect likely affects the probability of neighbouring defects. A canonical example is UiO-66, in which cluster vacancy defects assemble into synthetically tunable nanodomains[10,16,20], with the nanostructure affecting its mechanical properties[21,22]. Similarly, the disordered distribution of components in mixed-metal and/or mixed-linker MOFs, also known as multivariate (MTV) MOFs, may be correlated, which in turn determines their corresponding chemistry[23–26]. Incorporating low-symmetry components in MOF structures may also lead to orientational disorder. Examples are the linker 2,6-ndc in DUT-8(Ni)[27], the node in MOF c-(4,12)MTBC-$M_6$ (M=Zr, Hf)[28], and both the node and linker in TRUMOF-1—where orientational disorder of the bent linker 1,3-bdc is crucial to its unique aperiodic connectivity[29].

Similar to Zr-MOFs[28], lanthanide-MOFs (Ln-MOFs) are known to be structurally diverse, as a result of the highly versatile coordination numbers of the SBU[30]. Together with the unique physical properties of rare-earth metals, this variety in SBU geometry makes Ln-MOFs attractive as magnetocaloric materials[31], luminescent probes[32–34], and heterogeneous catalysts[35–38]. Given this structural diversity and range

[1]School of Chemistry, University of Birmingham, Birmingham, UK. [2]Institute for Integrated Cell-Material Sciences (WPI-iCeMS), Kyoto University, Kyoto, Japan. [3]ESRF, Grenoble, France. [4]ETH Zürich, Anorganische Funktionsmaterialien, Zürich, Switzerland. [5]Faculty of Geosciences, MARUM and MAPEX, University of Bremen, Bremen, Germany. [6]These authors contributed equally: Sarah L. Griffin, Emily G. Meekel, Johnathan M. Bulled. ✉e-mail: n.champness@bham.ac.uk

of potential applications, it is surprising that, to our knowledge, correlated disorder has not been investigated in Ln-MOFs.

## Results and discussion

In this context, we developed an interest in designing a Ln-MOF with nanostructured disorder. In our approach, we selected dysprosium (Dy) as the node component and a customized linker L designed for its flexibility and lower symmetry. These properties are known to introduce complexity in MOF design by increasing the degrees of freedom in linker orientation and/or strain, thereby enhancing the potential for correlated disorder[28,39,40]. As anticipated, the structure of the obtained MOF, to which we refer to as UoB-100(Dy), is governed by node disorder, which we discover to be nanostructured. In this communication, we report its synthesis, average structure and nanostructure, as derived from both Bragg scattering and 3D diffuse scattering analysis.

The linker, L, was designed to contain four carboxylate donors so as to encourage high connectivity but with a low-symmetry backbone capable of adopting multiple orientations [Fig. 1(a)]. The outer benzoate groups additionally impart a degree of flexibility, via

rotation about the benzoic acid-triazine bond. The synthesis of L was conducted via 'telescoped' condensation of pyridine-2,6-dicarbohydrazide with the appropriate 1,2-dicarbonyl (Scheme S1). The formation of related pyridinyl-1,2,4-triazine ligands has been previously reported[41,42]. The initial benzoin addition reaction between aldehydes was performed using thiamine hydrochloride[43] replacing the more toxic route that employs KCN as a catalyst[44]. For the conversion of the benzoin to a benzil, a Kornblum oxidation was chosen via an acid catalysed alcohol-halide substitution reaction. As before, other routes for this conversion are available, including routes using chlorine, nitric acid, and ammonium nitrate, but as with the first step the Kornblum oxidation was chosen under safety considerations[45–50]. Acid hydrolysis was used to convert the ester groups to carboxylic acid groups, avoiding the use of alkaline bases typically used in saponification reactions due to the propensity of benzil groups undergoing 1,2-rearrangments to form α-hydroxy-carboxylic acids[46]. The reactive bis(carboximidhydrazide)pyridine was obtained by reacting pyridine-2,6-dicarbonitrile with hydrazine monohydrate which was performed via a method adapted from Sagot et al.[47] The

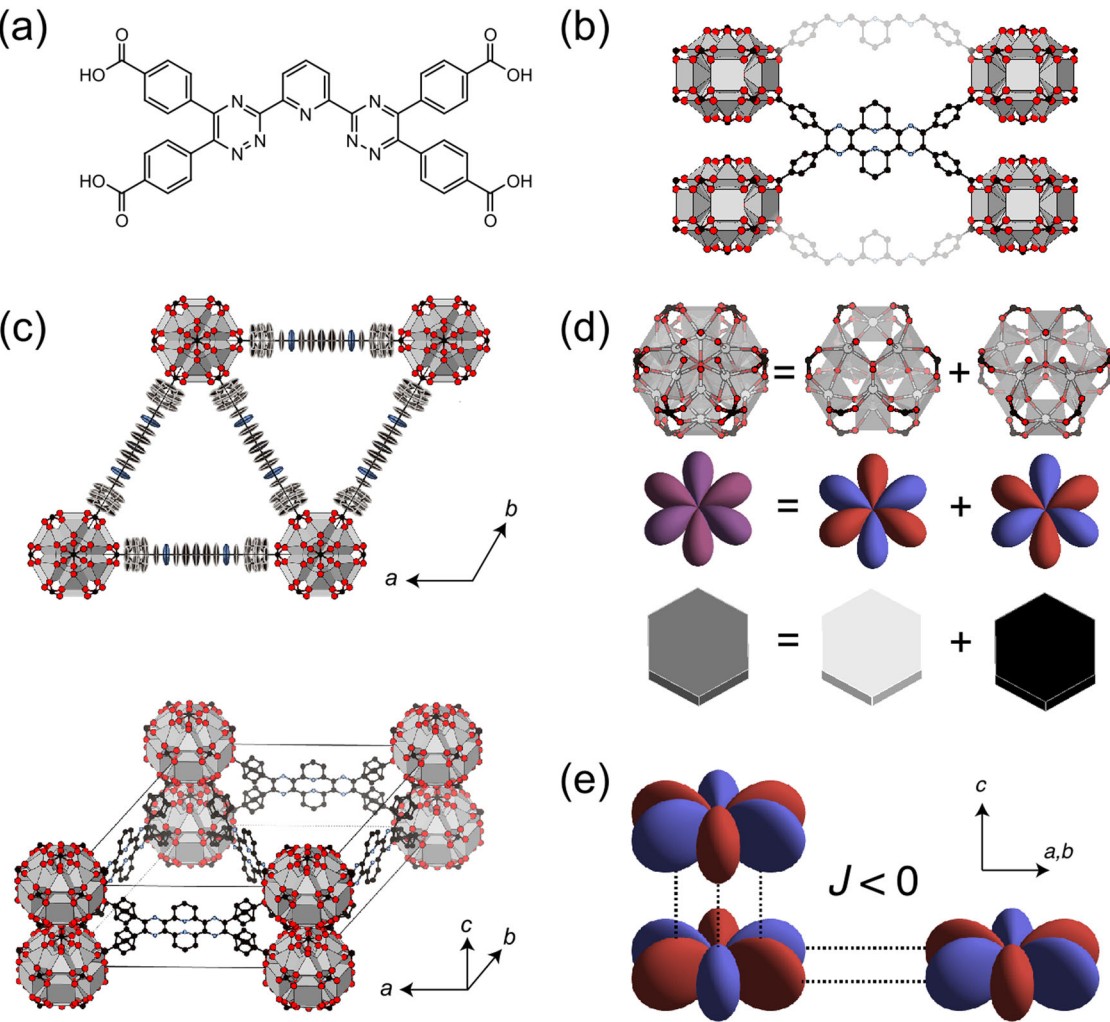

**Fig. 1 | Representations of the average structure of UoB-100(Dy). a** Molecular structure of linker L. **b** Arrangement of the linker in the average structure, illustrating the disordered up/down configuration and its connectivity to four nodes. Parts of two additional linkers are shown in transparent to highlight the connectivity of the nodes along *c*. Note the absence of coordinated components between the nodes, indicating their stabilization through electrostatic interactions. **c** Average unit cell of the crystal structure shown from above (*i.e.* along the *c*-axis) and the side. Colour scheme: Dy = gray polyhedra, C = black, O = red, N = blue. Linker in the top image is shown with ellipsoids at 50% probability. **d** The average structure of the Dy node (top left) can be understood as a superposition of two nonanuclear clusters at a different orientation (top right). Each cluster has a corresponding hexapole charge distribution (middle). For simplicity, we represent these hexapole orientations as black and white hexagonal tiles (bottom). **e** Antiferrohexapolar arrangements are electrostatically favourable, both between clusters stacked along the columns (*i.e.* along c) and within the same layer (i.e. in the *ab* plane.) (**c**, **d**: atom colours, Dy, light grey; C, black; N, blue; O, red).

final step in the synthesis of L proceeded via a method adapted from Tai et al.[42], and simply involved dissolving both reagents in a minimum of N,N'-dimethylformamide followed by heating at 80 °C overnight. All steps of the linker synthesis are relatively simple with minimal workup, using accessible and low hazard materials. The conformation of unbound L was determined by single crystal X-ray diffraction (SCXRD) (see S.I. for details).

UoB-100(Dy) was prepared as yellow hexagonal crystals from the reaction of $Dy(NO_3)_3.6H_2O$ with L in DMF using 2-fluorobenzoic acid as a modulator (see S.I. for details). SCXRD data were collected at a range of temperatures (100–300 K) and with either Cu-Kα or Mo-Kα radiation to ensure the best quality data for either diffuse scattering or Bragg diffraction. As a result, data collected at 100 K (Cu-Kα radiation) were used for diffuse scattering studies, and data collected at 250 K (Mo-Kα radiation) for average structure refinement (see S.I. for details).

The structure of UoB-100(Dy) comprises Dy-based SBUs bridged by twelve tetracarboxylate linkers, L [Fig. 1(b, c)]. The framework formed by UoB-100(Dy) contains $Dy_9$ SBUs that have been observed for other rare-earth (RE) MOFs with both tetracarboxylate[51–59] and tricarboxylate linkers[60–62]. Previous examples of MOFs containing the $RE_9$ SBU have been prepared using rigid linkers with little opportunity for alternative conformations. This class of MOFs, notably those with **shp** topology[51], has been widely studied and investigated for a variety of applications. Despite this interest, disorder of the SBU has been commonly observed[51–60] but no previous understanding of the correlation of the disorder has been proposed.

The crystal structure of UoB-100(Dy) is highly disordered, as can be derived from the many partially occupied atom sites in the average structure refinement [Fig. 1(b, c)]. We interpret the crystal structure by first considering the metal-containing secondary building unit (SBU), which is centred around the 1a Wyckoff site ($D_{6h}$ point symmetry). There are two symmetry-distinct Dy sites: 12o and 6m. Thus, there are 18 Dy atoms per SBU in total, although each of which has an occupancy of 0.5. In principle, when randomly selecting 9 out of 18 possible atom sites, there are $C(18, 9) = \frac{18!}{9!(18-9)!} = 48620$ possible node configurations with the correct number of Dy atoms. However, on closer inspection, only two of these configurations are chemically feasible: the closest contact between the nearest 12o Dy sites is 2.21 Å and the closest contact between 12o and 6m Dy sites is 2.66 Å. We reason that it is unlikely that pairs of Dy atoms are separated by such short distances and therefore exclude node geometries that include these small separations. Only two configurations satisfy these local rules, as shown in Fig. 1(d). We will go on to show experimentally that these nonanuclear clusters indeed exist in the local structure.

Besides the node disorder, it is clear from the average structure that the linker is disordered between two orientations: one where the linker points up along the c-axis, and one where it points down. Since the scattering from the linker disorder is expected to be much weaker than the significantly heavier Dy atoms in the nodes, it is difficult to resolve correlations in linker orientation experimentally. For this reason, we solely focus on the dominant node disorder in our initial model.

The presence of the disordered nodes raises the question: how are these nodes distributed in space? To answer this question, we need to understand the interactions between the nodes. In our approach, we represent the local configurations of the nodes using hexapole charge distributions, preserving their $D_{3h}$ point symmetry [Fig. 1(d)]. While a 60° rotation of these hexapoles is mathematically equivalent to the flipping of Ising spins, the hexapolar representations provide a clearer picture of the charge distribution of the nodes and, in turn, the electrostatic interactions between them (see S.I. for details). This disordered arrangement of multipoles draws comparison to magnetic multipole liquids $Ce_2Sn_2O_7$ and $URu_2Si_2$[63–66], as well as the assemblies of molecular multipoles in barocaloric plastic crystalline phases and

molecular perovskites[67,68]. In each case, understanding the correlated disorder present via diffuse scattering and modelling approaches was crucial to understanding the systems.

Based on electrostatic arguments, it is expected that antiferrohexapolar interactions are favoured between such neighbouring nodes, both along the c-axis and within the ab-plane, as shown in Fig. 1(e). We can estimate an upper bound of the strength of these electrostatic interactions by assigning each atom its full formal charge (see S.I.). Our calculations reveal that the interactions along the c-axis (4.111 kJ mol⁻¹) are significantly stronger than those within the ab-plane (0.0674 kJ mol⁻¹). For reference, the thermal energy at the crystallization temperature is approximately 2.479 kJ mol⁻¹. These results suggest that the electrostatics are likely to govern the arrangement of nodes along the c-axis, while playing a minimal role in driving ordering within the ab-plane. We will go on to show that this is indeed the case experimentally.

To test our hypothesis and analyse the correlations of the node disorder experimentally, we turn to diffuse scattering (DS). A full description of the data collection, treatment, and 3D-ΔPDF extraction can be found in the S.I. We observe significant DS in the 0kl and h0l layers, structured into sharp planes perpendicular to l [Fig. 2(a-b)]. The linewidth of these planes is resolution-limited (FWHM < 0.03 reciprocal lattice units (r.l.u.)), implying a correlation length along the c direction of $\zeta_x > 14$ nm. In addition, there are maxima around $[h, k, l+\frac{1}{2}]$: $h, k, l \in \mathbb{Z}$ within these planes of DS, corresponding to a shorter-range correlation of $\zeta_x = 3.9$ nm. Accordingly, we observe strong features in the 0yz and x0z layers of the 3D-ΔPDF, with alternating signs along the z direction and the same sign along the x and y directions [Fig. 2(c)]. We first note that the features of alternating signs along z correspond to the observed planes of DS, using the condition $l = n + \frac{1}{2}$: $n \in \mathbb{Z}$. Therefore, antiferrohexapolar correlations apply along the z direction. Second, we derive the shorter-range correlations in the xy plane to be ferrohexapolar, based on the same sign of the features within these planes [Fig. 2(c-d)]. Thus, while our electrostatic arguments correctly predicted antiferrohexapolar correlations along z, they cannot explain the ferrohexapolar interactions correlations present in the xy plane.

Notably, the DS features are rather localized. For this reason, we integrated the additional diffuse intensity in the same way we would for Bragg scattering (see S.I. for details). This combined Bragg+DS refinement gives the local structure within nanodomains, on the length-scale discussed in the previous paragraph[69], in the same way as for the small-box modelling performed on powder PDF data[70,71]. Since the DS peaks are located in the half-integer layers, the lattice doubles along the c-direction, corresponding to the space group $P6_3/mmc$; a maximal isomorphic subgroup of the parent P6/mmm[72]. In this supercell, we first find that the linker is disordered over the up/down configurations, as previously observed in the average structural refinement [Fig. 1(b, c)]. Second, the metal cluster is also disordered over two configurations, with four unique Dy atoms in the asymmetric unit instead of two. We constrained the occupancy of these Dy atoms related by a change of SBU orientation to sum to one; there are two such pairs [Fig. 3]. These occupancies refined to 0.635(3):0.365(3) for one pair (Dy1:Dy2) and 0.662(4):0.338(4) for the second pair (Dy3:Dy4). The majority sites (Dy1, Dy3) are shown in the right panel of [Fig. 3] with the same geometry as our prediction in Fig. 1(d). Moreover, the c-glide plane perpendicular to c dictates that the orientation of the SBUs must change from one layer to the next, entirely consistent with our hypothesis of local antiferrohexapolar correlations along c.

To better understand the correlated disorder governing the system, we turn to a microscopic model of the SBU orientations using a Monte Carlo (MC) approach. This method involves generating a representative sample of the statistical distribution of disordered super-cells. It is evident from the 3D-ΔPDF that the node orientations are not random, hence, a method of including these correlations in our

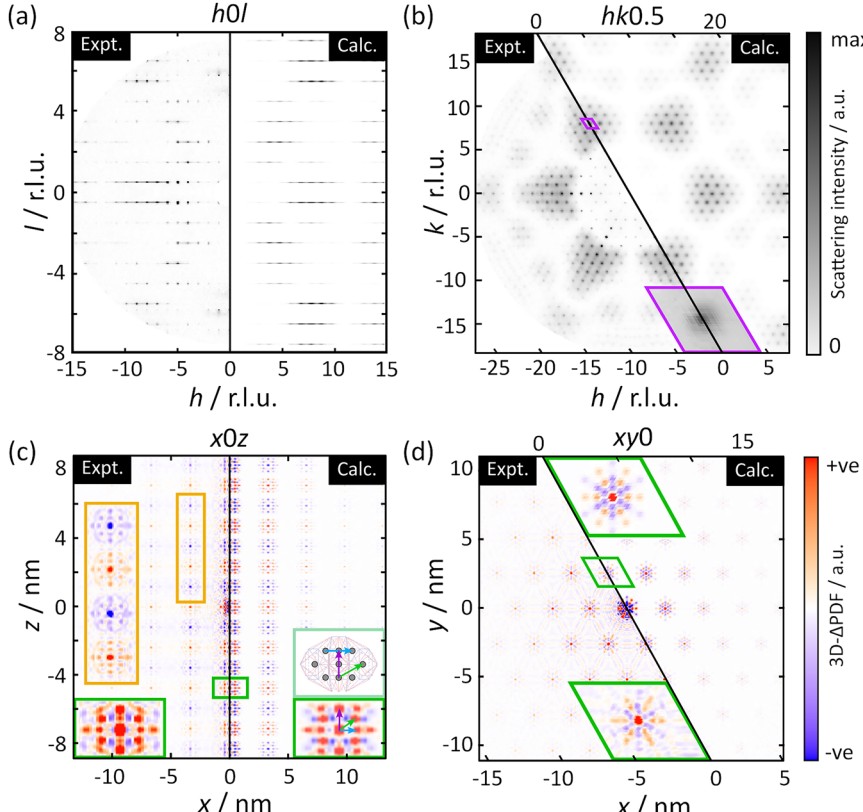

**Fig. 2 | Comparison of DS and 3D-ΔPDF obtained from experiment (left of each pane) and the best fit of the Monte Carlo model described in the text (right of each pane). a** $h0l$ and (**b**) $hk\frac{1}{2}$ planes of diffuse scattering. Note that the scattering observed in the $l = 5$, 6 planes likely originates from thermal diffuse scattering condensing around the Bragg peaks. The purple rhombuses in (**b**) show the region bounded by $[70\frac{1}{2}] \pm [\frac{1}{2}\frac{1}{2}0]$ used for fitting, and the inset shows this region enlarged. The (**c**) $x0z$ plane and (**d**) $xy0$ planes of the 3D-ΔPDF. Insets in green highlight the regions around r = [100] and [005] cell-vectors for (**c**) and (**d**), respectively. In (**c**), an additional inset is given in light-green to highlight the relationship between the metal cluster Dy atom distances and the features observed in the 3D-ΔPDF. The inset in orange highlights the alternating sign of the features along $z$.

simulation is necessary. Given that the correlation length along $c$ is long, we make the approximation that the disorder is present only in the $ab$ plane, with long-range antiferrohexapolar correlations persisting in the $c$ direction. Within the plane, the tendency for ferrohexapolar interactions can be encoded in the Hamiltonian,

$$E_{MC} = J_{\perp} \sum_{\langle ij \rangle} S_i S_j \tag{1}$$

where $S_j = \pm 1$ represents which of the two states in Fig. 1(b) is present at site j in the lattice; $J_{\perp}$ is the coupling constant of nearest neighbours ((denoted $\langle ij \rangle$, with $i < j$). Since the coupling in the ab plane is ferrohexapolar ($J_{\perp} > 0$), neighbours favour the same spin state. We parameterise the value of $J_{\perp}$ by fitting simulated data to experimental scattering using a small, representative area of intense DS (shown in purple in Fig. 2(b)). The procedure for calculating DS from our model and fitting to data is detailed in the S.I. In this way, correlations present in the 2048 atoms of our supercells can be encoded with a single parameter: $J_{\perp} / T_{eff} = 0.1361$.

The DS and 3D-ΔPDF cuts calculated from our final MC model are shown in Fig. 2, demonstrating exceptional agreement with experimental data. Notably, the agreement of the features corresponding to the node-node correlation functions (shown in the insets of [Fig. 2(c-d)]) confirms that we have correctly identified the two SBU orientations [Fig. 1(d)]. Furthermore, the agreement of the extent of these correlations in real-space shows that we have correctly reproduced the nanoscale correlations of these SBU orientations.

A simplified representation of the local structure of UoB-100(Dy) is given in Fig. 4(a). We clearly observe two features that exist on the nanoscale: the extremely long-range correlations along $c$, and the shorter-range ($\zeta_x \sim 5$ nm) correlations in the $ac$ plane, of antiferrohexapolar and ferrohexapolar nature, respectively.

In our analysis, we have deliberately neglected the disorder of the linker, because its effect on the DS signal is weak. We see some small deviations between the model and experiment at low-Q in Fig. 2(b), which may be attributable to linker ordering or relaxation. In the 3D-ΔPDF [Fig. 2(d)], there is some weak evidence for correlations between the internode distances, which emerges due to correlations between the node orientation and the linker. Understanding these correlations is complex, as they depend on the orientation of both the node and the linker. Instead of modelling this disorder and comparing to data, we take a theoretical approach to understanding this linker relaxation in response to the orientations of the SBUs it is bound to. More specifically, we use the experimentally derived oxygen positions for the ferro- and antiferrohexapolar node orientations to constrain a density functional theory (DFT) optimisation of the linker geometry. The details of this simulation are given in the S.I.

Our DFT results show two things [Fig. 4(b,c)]. First, in the ferrohexapolar case, the linker is allowed to relax into the pore, bending away from the $c$-direction by 28.5°. In the antiferrohexapolar case, however, symmetry dictates that the linker must be aligned along $c$. The large angular relaxation in the ferrohexapolar configuration partially explains the high degree of linker disorder, as reflected by the anisotropic displacement parameters obtained in the average structure refinement [Fig. 1(c)]. Second, with a difference in relaxed energy

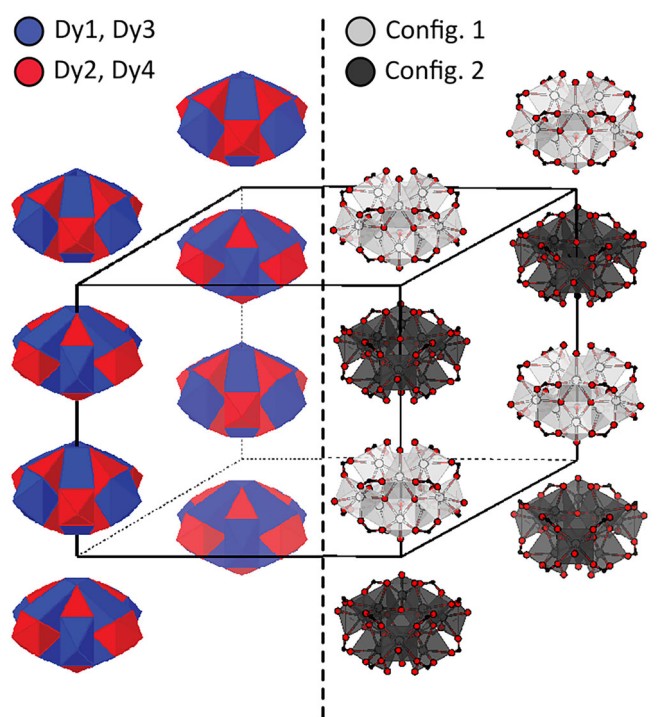

**Fig. 3 | Depiction of the SBU arrangement in the refined local structure.** The SBU arrangement in the refined local structure, derived from the 'Bragg+Diffuse' refinement described in the text with the location of the Dy atom pairs are shown in the left panel and the configurations of the majority sites in the right panel, alternating between the two configurations shown in black and white. (atom colours, Dy −light grey; C, black; O red).

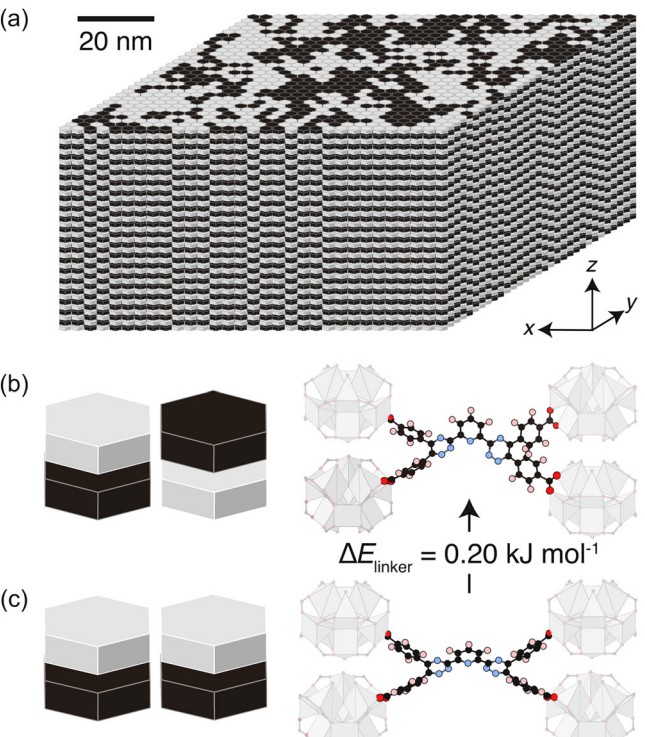

**Fig. 4 | The nanostructure of UoB-100(Dy). a** Representation of one of the nanostructures of UoB-100(Dy), as produced by the MC procedure detailed in the text. The colour of the hexagons represents the SBU orientations as shown in Fig. 1(d). Note that there is order along $c$, and nanoscale order in the $ab$ plane. Black/ white blocks represent single SBUs in different orientations, as shown in Fig. 1, which are ordered in our model along the c-direction. **b** The structure which represents an antiferrohexapolar arrangement of the SBUs in the plane and DFT-relaxed linker arrangement constrained by these adjacent node geometries. **c** The same arrangement for ferrohexapolar neighbours. The constrained linker geometry optimizes with a lower energy, indicating that this ferrohexapolar node arrangement leads to less strain. (note, DFT calculations were performed solely on the linker.) (**b**, **c**: atom colours, C, black; N, blue; O, red; H, pink).

of $\Delta E = 0.20$ kJ mol$^{-1}$, the linker is considerably more strained in the antiferrohexapolar configuration than it is in the ferrohexapolar configuration, likely because the latter allows the linker to relax in the pore. While the energy difference itself is small when compared to the thermal energy at the temperature of crystallization ($J \sim 0.5$ kJ mol$^{-1}$, $kT = 2.479$ kJ mol$^{-1}$: $J < <kT$), the long correlation length along $c$ indicates that many of these interactions contribute additively to the overall interaction energy between chains. That being said, the relatively small energy difference between conformations may explain why long-range ferrohexapolar order is suppressed. Overall, these observations explain the preference for ferrohexapolar correlations between the SBUs, as seen experimentally. As the experimental data confirmed our hypothesis of electrostatic interactions governing the node arrangement along $c$, we did not include the ferrohexapolar arrangement of stacked nodes in our DFT calculations. However, we propose that linker steric effects, particularly to prevent clashes of the linker benzene groups, further favour the antiferrohexapolar alignment.

There are multiple ways in which the observed nanodomains of UoB-100(Dy) may affect its physical function. First, we note the potential impact of the linker bending, as it occurs within the ferrohexapolar nanodomains, but is absent along their edges. Since the linker bending affects the size of its pore windows, UoB-100(Dy) may exhibit disorder-dependent behaviour concerning gas storage and transport. Or, if the linker bending is a mode of mechanical flexibility, UoB-100(Dy) could show an enhanced elastic stability through combinatorial mechanics, as reported previously for TRUMOF-1[73]. Moreover, because the linker possesses a cavity with the ability to coordinate to an added guest component, the possibility of modifying the bending arises. This prospect of nanodomain modification is what sets UoB-100(Dy) apart from other nanostructured disordered systems, such as UiO-66[16], relaxor ferroelectrics[8], high-entropy alloys[48], or

charge density wave compounds[49,50]. Finally, as Dy exhibits strong magnetic behaviour[74,75], it is likely that the nanostructured disorder affects the magnetic structure and behaviour of UoB-100(Dy). It may be interesting probing UoB-100(Dy) for the magnetocaloric effect, which has been reported for Dy-coordination polymers such as DyOHCO$_3$ and Ln(HCO$_2$)(C$_2$O$_4$)[76,77].

More generally, this study shows the importance of local-structure analysis in MOF chemistry. Often, signs of non-average effects are ignored, leading to the incorrect assignment of components, such as the metal cluster geometry in Zr-MOFs[28]. Likewise, structures with similar cluster geometries to UoB-100(Dy) may have been observed before, but difficulty in structure analysis prevented the data from being disclosed. By designing systems with similar multipole order, one may be able create analogues of the frustrated magnetic multipole liquids[63–66]. As we demonstrate for UoB-100(Dy), both DS analysis and multimodal modelling are key in deciphering such cases. In parallel with structural disorder analysis, the use of disorder as a tool in framework design becomes increasingly critical. This targeted design is a significant challenge that must not be overlooked, as it involves navigating a large configurational energy landscape and therefore some (synthetic) trial and error. In our case, the approach of combining a flexible linker with a rare-earth metal proved to be efficient. The presence of hexapolar charge distributions in the Dy$_9$ clusters also hints at a role for local electrostatic interactions in influencing local correlations. It will be beneficial to explore the potential role of

such electrostatic interactions for other SBU configurations. There is no reason to think that the extension to other rare earth elements, or indeed other metals, is not feasible, although it might affect the disordered structure. Ultimately, solving the nanostructured node disorder in UoB-100(Dy) opens a new avenue of exploring the control thereof, be it through the addition of guest components, changing metal type, or other synthetic parameters.

## Methods

All the reagents used were purchased from commercial suppliers and used without further purification. NMR spectra were recorded on a Bruker AVANCE NEO 400 MHz spectrometer and referenced to residual solvent peaks, unless otherwise stated. Deuterated solvents were used as specified. Details of the synthetic pathway to linker L are given in supplementary information.

### Synthesis

Single crystals of UoB-100(Dy) were grown in the following manner. Linker *L* (7.0 mg, 9 μmol), dysprosium (III) nitrate hexahydrate (13.0 mg, 28 μmol), and 2-fluorobenzoic acid (0.60 g, 1.4 mmol) were added to a 25 ml Schott bottle, followed by *N*,*N*-dimethylformamide (3 ml). The reaction jar was sonicated until all solids fully dissolved, followed by the addition of acetic acid (0.7 mL). The bottle was once again sonicated prior to placing in the oven at 120 °C for six days. During this period, yellow hexagonal crystals were formed. Upon cooling the crystals were isolated and washed with *N*,*N*-dimethylformamide.

### Average structure determination and refinement

Single-crystal diffraction data for average structure determination were collected on a Rigaku XtaLAB Synergy-S X-ray diffractometer equipped with a HyPix-6000 hybrid photon counting detector and a Mo microfocus source. A suitable crystal was isolated and mounted on a MiTeGen loop in a droplet of a perfluoropolyether oil.

A series of omega scans were conducted (calculated by the automatic procedure from the software manufacturer) to ensure sufficient redundancy and completeness of reflections, using an oscillation of 0.5° per frame. During data collection, the temperature was controlled by an Oxford CryoStream at a value of 250(1) K (reported by the Software interface). Data reduction was conducted by using the software CrysAlisPro[78]. A clean, single lattice was identified from the harvested peaks. While no twinning/multicrystal was observed, systematically weaker superstructure reflections were spotted (see Supplementary information Fig. S2), and excluded from the integration as assigned to local structure effects. Intensities integration was based on a primitive hexagonal unit cell, while absorption correction was conducted by a multiscan approach as implemented in CrysAlisPro.

Crystal structure solution and refinement were conducted by using the Olex2 software ver. 1.5[79]. Structure solution was readily found by the programme ShelXT[80] used within the Olex2 user interface, already revealing the split positions of the Dy cluster and the coordinating parts of the linkers. The remaining atoms were assigned manually and a series of ShelXL restraints were used to drive the structure towards a reasonable molecular geometry, which also agreed with the underlying electron density (see Supplementary information Figs. S3, S4). The final refinement was conducted by the least square algorithm of the ShelXL structure refinement programme[81]. Further details are available in the supporting information.

Crystal Data for $C_{124}H_{97.96}Dy_9N_{15}O_{50}$ ($M = 4060.63$ g/mol): hexagonal, space group $P6/mmm$ (no. 191), $a = b = 28.6350(5)$ Å, $c = 12.5830(2)$ Å, $V = 8935.3(3)$ Å$^3$, $Z = 1$, $T = 250(1)$ K, μ(Mo-Kα) = 1.891 mm$^{-1}$, $D_{calc} = 0.755$ g/cm$^3$, 137655 reflections measured ($3.63 \leq 2\Theta \leq 56.5582°$), 4221 unique ($R_{int} = 0.0584$, $R_{sigma} = 0.0161$) which were used in all calculations. The final $R_1$ was 0.0871 (I > 2σ(I)) and $wR_2$ was 0.3172 (all data). Details of dealing with disorder and other refinements are described in the corresponding deposited cif, CCDC

2368155, but are also described in the Supplementary Information file for convenience.

For the supercell structure determination and refinement, the same dataset was used as for the average structure determination. Data reduction was conducted by using the software CrysAlisPro[78], this time including the diffuse scattering at $n =$ half integers for the peak integration (see supplementary information Fig. S2). Intensities integration was based on a primitive hexagonal unit cell, while absorption correction was conducted by a multiscan approach as implemented in CrysAlisPro. Crystal structure solution and refinement was conducted by using the Olex2 software ver. 1.5[79]. Structure solution in $P6_3/mmc$ was achieved by the programme ShelXL[81] used within the Olex2 user interface, revealing two Dy atoms of the metal cluster. The remaining atoms were assigned manually and a series of ShelXL restraints were used to drive the structure towards a reasonable molecular geometry (as derived from the average structure solution), which also agreed with the underlying electron density. The final refinement was conducted by the least square algorithm of the ShelXL structure refinement programme[81]. Further details of dealing with disorder and other refinement are described in the Supplementary Information file.

### Diffuse scattering measurement and analysis

The single-crystal X-ray diffuse scattering measurement of a Dy-MOF crystal was collected on a RigakuXtaLAB Synergy diffractometer fitted with a HyPix-6000 detector. The dataset was collected under Cu radiation (λ = 1.5406 Å). Crystals were mounted on a MiTeGen loop using perfluoropolyether oil as a cryoprotectant. An exposure time of 10 s was used to detect diffuse temperatures at 100 K. The measurement involved a full 360° φ-scan with 0.2° rotation/frame carried out in a single run. Raw data can be found at https://doi.org/10.5281/zenodo.14269933.

### 3D reciprocal space reconstruction

CrysAlisPro[82] was used for indexing, determination and refinement of the orientation matrix. For the 3D reciprocal space reconstruction, we utilized the software Meerkat[83]. As we observed a slight movement of the sample during the measurement, we utilized XDS[84] to re-refine the orientation matrix after every 60° φ-rotation, which corresponds to 300 frames. The scattering data were then reconstructed on a three-dimensional grid defined by $-30 \leq h, k, l \leq +30$ with voxel sizes of $\Delta h = \Delta k = \Delta l = 0.05$ r.l.u., resulting in an array of $1201 \times 1201 \times 1201$ voxels. To improve statistics and cover missing parts of reciprocal space, the data were subsequently averaged for 6/mmm Laue symmetry using a custom Python script. Data treatment for the experimental 3D-ΔPDF generation consisted of Bragg peak removal and the subtraction of a constant background. Bragg peak removal was performed using a custom Python script that punches a pre-defined area around the Bragg peak and interpolates the missing intensities. The Fast Fourier Transform (FFT) algorithm as implemented in Meerkat[83] was used to obtain the 3D-ΔPDFs shown in Fig. 2.

### Monte Carlo simulations

Monte Carlo (MC) simulations were carried out using the Metropolis algorithm[85]. The MC simulation was carried out using a custom code related to that used in ref. 86. All simulations were carried out on $24 \times 24 \times 1$ supercells. To reach the ground state we used the approach of simulated annealing, cooling from $T = 15$ to $5J_\perp$ on a logarithmically spaced grid of 23 points. For each temperature, the algorithm measures the decorrelation time $n_d$, equilibrates for $10n_d$, and progresses to measure 80 samples with $2n_d$ moves per sample. In this way, we ensure ergodicity in the simulations.

### Calculation of diffuse scattering

Calculating the diffuse scattering required decorating the $24 \times 24 \times 1$ supercells with the cluster geometries presented in

Tables S2 and S3 (see Supplementary Information), and using the software SCATTY (which exploits fast-Fourier transform) to calculate various planes[87]. To include the antiferrohexapolar order along the $c$ direction, it was necessary to extend the $24J_\perp/T$ 24×1 supercells to $24 \times 24 \times 2$, using the rule $S(x,y,z+1) = -S(x,y,z)$ to calculate the spins at all sites $S(x,y,z)$ based on our simulated values for $S(x,y,1)$. SCATTY was used to calculate the $hk0.5$ plane, with no Lanczos resampling. The maximum in-plane resolution of the simulations was given by the size of the supercell: 1/24 r.l.u. However, the $24 \times 24 \times 2$ supercells did not give sufficient resolution along $c$ to simulate the $h0l$ plane. To do so, we therefore used the convolution theorem, noting the periodicity along the $c$-axis to extend the resolution arbitrarily in this direction. The Fourier transform of the periodic lattice along the $c$-direction is a set of planes perpendicular to $c$, which were multiplied by the simulated $h0l$ planes to give the data plotted in Fig. 2a. The width of these planes is ultimately arbitrary as they are resolution limited in the experiment, but a width was chosen that makes them visible.

### Parameterization of the model

The Hamiltonian was parameterized by fitting to the scattering data. As noted in the text and by Schmidt and Neder[88], in the case of disorder between two orientations of a molecule, only one Brillouin zone is required to encode all of the correlations present. For this reason, a representative region of reciprocal space was chosen to parametrize the model, bounded by $[7\ 0\ 0.5] \pm [0.5\ 0.5\ 0]$

Since the scattering considered is localized in the $hk0.5$ plane, to remove the background, a region 0.05 r.l.u. above and below the plane were subtracted to give the experimental diffuse scattering in this region, shown in Fig. S5a (see Supplementary Information). By simulated annealing, a number of different values of $J_\perp/T$ were sampled, and the corresponding diffuse scattering (which becomes sharper at high-coupling and diffuse at low-coupling) was plotted. In this way, we could fit the experimental sharpness of this feature. Normalization of these two datasets was achieved by matching the maximum intensity within the region of reciprocal space. The resulting residual is shown in Fig. S5a (see Supplementary Information), with a goodness-of-fit as a function of our $J_\perp/T$ parameter,

$$\chi^2 = \sum_{h,k}(I^{\exp}(h,k,0.5) - I^{\text{calc}}(h,k,0.5))^2 \tag{2}$$

This whole procedure was repeated 4 times, and a distribution of $\chi^2$ values across these samples gave the mean and $2\sigma$-uncertainty plotted in Figure S7b (see Supplementary Information). With this approach, best fit of the model-parameter of $J = 0.144(12)\ T_{\text{eff}}$ was determined.

### Calculation of model 3D-ΔPDF

The full volume of reciprocal space was calculated from the best-fit configurations in the same way as for the other scattering data, using SCATTY[87]. A max-$hkl$ of 20 r.l.u in each direction was used, and a $401 \times 401 \times 81$ grid of reciprocal lattice points, giving the maximum resolution of the $24 \times 24 \times 2$ supercells in the $c$-direction. In this way, the data plotted in Fig. 2d were obtained. To broaden the simulation to match the real-space experimental resolution, a gaussian convolution of a width of 0.0833 r.l.u is applied, as matched by eye to give the best agreement with experiment. Rather than using the multiplication approach used to calculate the diffuse scattering, the periodicity in the $c$-direction was directly applied to obtain the 3D-ΔPDF presented in Fig. 2d.

### Quantum mechanical calculations

DFT calculations were carried out using the software package ORCA[89], using the PBE functional[89], and the def2-svp basis set[90]. By default, the TightSCF convergence of the SCF cycles is used[91], and the NormalOpt criterial is used for the geometry optimization. Two symmetry constraints were used: for the antiferro configuration, the starting model had $C_2$ symmetry, while for the ferro configuration, we started with the point group $C_s$. Further details and discussion can be found in the Supplementary Information file.

## Data availability

The authors declare that all characterisation data generated in this study are provided in the Supplementary Information or within the main manuscript. The X-ray crystallographic coordinates for structures reported in this study have been deposited at the Cambridge Crystallographic Data Centre (CCDC), under deposition numbers 2368155 and 2370912. These data can be obtained free of charge from The Cambridge Crystallographic Data Centre via www.ccdc.cam.ac.uk/data_request/cif. Crystallographic information files (CIF). Raw data can be found at DOI: 10.5281/zenodo.14269933.

## Code availability

All custom code used in this study was developed using widely available algorithms. Copies of the actual code used can be obtained upon request. See also reference S7 in the Supplementary Information.

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

## Acknowledgements

N.R.C. gratefully acknowledges support from the UK Engineering and Physical Sciences Research Council (EP/S002995/2). E.G.M. thanks the Japan Society for the Promotion of Science (JSPS) for support. The Euler scientific computing centre of ETH Zurich is acknowledged for providing its useful resources.

## Author contributions

S.L.G., E.G.M. and J.M.B. contributed equally to the overall experimental aims of the study. The synthesis of materials and other experimental studies was conducted by S.L.G. and A.W.L. S.C. and E.G.M. performed analysis of the SCXRD data, leading to modelling of the average structure, curated the reciprocal space reconstruction, correction and 3D PDF extraction. E.G.M., J.M.B. and E.M.S. evaluated the correlated disorder and performed all related modelling. The project was conceived and supervised by N.R.C., E.G.M. and J.M.B. wrote the initial draft of the manuscript. All authors discussed the results, contributed to and have given approval to the final version of the manuscript.

## Competing interests

The authors declare no competing interests.
