## [Transparent Peer Review file · Nature Communications]

A lanthanide MOF with nanostructured node disorder

Corresponding Author: Professor Neil Champness

Version 0:

Reviewer comments:

Reviewer #1

(Remarks to the Author)

In this paper, Griffin et al report the structure of a new Dy based MOF. They prove that the 'Dy18' cluster observed from single crystal diffraction is in fact the disordered super position of two 'Dy9' clusters, as previously assumed. Further, they show that the Dy9 clusters alternate in orientation along the c axis, but are only weakly correlated along the a and b directions. They use state of the art 3D Delta PDF methods to determine the structure, combined with monte carlo modelling to estimate interactions. This is a very neat study showing that there is significant additional structure complexity present in this family of MOFs, and should be expected. I think overall this paper will be a very valuable contribution to the literature.

There are some issues the author should address:

- The authors mention some of the other MOFs that feature disorder of an analogous cluster. They don't mention a number of other MOFs highlighted in Ref 54, principally those by the Eddaoudi group, or indeed disorder found in the NU family of 'shp' topology MOFs. This might help make the generality of the discovery clearer.

- It took some effort to figure out the structure of the material from the main text: the ligand is never named and never shown clearly. I would suggest adding both. The composition of the cluster is also not specified clearly in the text. Further, the authors show that fluoride is likely in their MOF (as per Balkus <https://doi.org/10.1021/jacs.1c10493>), but there's no evidence of this in their reported composition.

- What is the convention for J w.r.t. single and double counting of pairs. It is unclear from the text and SI.

- The authors rely on electrostatics and an 'hexapole' to justify the ordering along the c axis. This seems unlikely to be the real driving force, particularly as the authors show large changes in the molecular geometry of the ligand between orientations which lead to some energy differences. The authors also don't appear to relate the MC J with their DFT J. Using the synthesis temperature as 'T' gives I think $\sim 0.5 \text{ kJ mol}^{-1}$, which is surprisingly close to the estimate from DFT. Perhaps this sort of analysis could be included.

- What consequence does the hexapole have in the structure and modelling? It is not clear what additional insight this complexity brings. It also seems the authors model it as an Ising degree of freedom (and perhaps also refer to it as spin up and down in the SI?). This would probably be clearer to audiences less familiar with such high order multipoles.

- Could the authors estimate the energetic difference due to electrostatics and compare it to the DFT & MC derived energy?

- Fig 2 would benefit from the cluster being shown on the same scale as the PDF.

- "While the energy difference itself is not small compared to the thermal energy at the temperature of crystallization" This discussion is a little confusing: perhaps it could be rephrased.

I don't understand Figure 3. Why are there 4 blocks in each stack when there are only 2 clusters in each stack. Figure 3 also implies the authors include the clusters in their DFT calculations, which they do not.

Minor issues:

- "Although such structural disorder can be random, it is more often correlated.^{6,7}"

Correlated disorder is certainly common, but I don't think there is much evidence it is the majority of disordered systems. The references don't support it.

- Reference 21 could be augmented by recent work from Chapman et al in JACS.

- "In principle, there are $C(18, 9) = 48620$ possible nodes with this correct number of Dy atoms"

Is this right? It seems with the high site symmetry there are considerably less.

- The authors say all relevant data is in the SI. I would strongly encourage them to be more expansive in the included data. In particular, the authors only include 2 cuts of their PDF data and total scattering in the main paper. Please can the authors include the DFT results as files, rather than just PDF tables as they are hard to parse.

- I would also encourage them to include the CIFs with the SI, as it saves a reader/reviewer going to the CSD to download them

- What does spin up and spin down mean in the cluster geometry section of the SI?

- Typos: 0.20 kJ mol⁻¹

Reviewer #2

(Remarks to the Author)

The work by Champness and collaborators is a valuable contribution to the concept of correlated disorder in molecular architectures. The concept of defects is extensively used to manipulate physical characteristics (such as mechanical properties, thermal stability, and porosity) and chemical properties (such as the generation of open metal sites and reactivity) in Metal-Organic Frameworks (MOFs). However, many studies lack an advanced analysis of how these defects can organize periodically within the solid. This is precisely the focus of this article, which goes beyond complementing previous studies (such as those exploring this phenomenon with UiO-66 using X-ray scattering by Goodwin) by extending this analysis to new materials (RE-MOFs) and introducing the use of 3D diffuse scattering and computational modeling as tools for analyzing this phenomenon in single crystals, thereby generating descriptive information on the distribution of correlated nanodomains within the crystal.

This study has significant potential to attract attention and citations from colleagues working in the design and modification of MOFs and introduces a novel tool that will greatly benefit a smaller yet growing community interested in the structural analysis of defects in molecular solids.

Suggestions for clarification:

- In the discussion on the linker, the authors describe its low symmetry as a design element to facilitate defect formation (by orientation?). I would recommend elaborating more clearly on the relationship between the connectivity/flexibility/symmetry of the linker and its effect on the generation of correlated defects.
- Figure 1 does not clearly help to visualize the characteristics of the cluster and the linker. I would suggest adding a sub-figure to clearly illustrate this before introducing thermal factors or crystallographic disorder.
- The text references the formation of up to 48,620 possible nodes with different positioning of Dy atoms in the Dy₉ cluster. How was this number calculated? Please clarify.
- Prior to the discussion of the scattering data, there is an assumption regarding the possible distributions of Dy₉ nodes in space, driven by the presence of ferro- or antiferro-hexapolar charge distributions. In my opinion, it would be beneficial to delve deeper into the origin of these local interactions and whether these electrostatic interactions might extend to other clusters with different symmetries commonly used in MOF design.

I do not consider this essential for publication; however, the study and its impact would greatly benefit from demonstrating how the distribution of antiferro- or ferro-hexapolar nanostructures might be modified with specific experimental parameters. Collecting scattering patterns at variable temperatures or in the presence of electric/magnetic fields could reveal a modulation in the distribution of nanodomains, which in other cases as UiO-type materials would only be accessible through synthetic variations.

Reviewer #3

(Remarks to the Author)

The authors present the synthesis and structure solution of MOF UoB-100(Dy). The unusual part of this paper is the use of diffuse scattering to gain insight into the local structure of the MOF. The authors present single crystal diffuse scattering and Monte Carlo modelling to understand its local structure. The model fits the experiment remarkably well, and I have no doubt that it is correct. I recommend the paper for submission.

One particular feature I would like to highlight is that the authors use an efficient separation of diffuse scattering, first performing the determination of Ising interaction constants from a small portion of reciprocal space. Such separation saves

time, and since diffuse scattering appears in diffraction of many MOFs other groups can benefit from this strategy in the future.

I have a few questions and comments:

Could the authors add the enlarged diagram of the linker to the Figure 1? (same as on S1) Currently it is too small to understand it.

Figure 2a shows diffuse scattering with punched Bragg peaks from the main structure. Interestingly**,** in some layers the punched Bragg peaks don't align with the plane they are expected to be (see attached image with the red line of hk6 plane). Are those peaks from a secondary phase?

Figure 2b shows a good fit of diffuse model to experiment. However, experimental scattering has peaks at low reciprocal space vector, while the model does not. Could the authors discuss potential source of this discrepancy?

"Based on electrostatic arguments, it is expected that 'antiferro-hexapolar' interactions are favoured between such neighbouring node". From the experiment you have shown that in the ab plane the electrostatic argument is invalid, and it is the steric strain of the ligand which defines the sign of the nearest neighbour interaction. Could the authors discuss their expectations for preferred stacking along the c axis, based on the steric argument? Is there any obvious reason why ligands would prefer antiferro-ordering along c axis?

While it is commendable that the authors have done the diffuse scattering modelling, since diffuse scattering is well concentrated in half-layer peaks, did the authors try to integrate those diffuse ****peaks as Bragg peaks, and use those new peaks to refine the supercell of the structure? If successful, this can provide experimental insight into the relaxation of ligands.

"these nodes assemble into a complex and novel nanodomain structure" - **This appears to be** an imprecise statement. First, the current paper doesn't give any direct access to the *domain* structure, only to local correlations which suggest that such domain structure should probably exist with anisotropic shape of the domains. I suggest removing this sentence.

Typos:

Page 1: Cu_x Mn (should it be Cu_x Mn_{1-x} ?)

in SI some references are missing, they are just indicated as (REF)

Data availability:

" The X-ray crystallographic coordinates for structures reported in this study have been deposited at the Cambridge Crystallographic Data Centre (CCDC), under deposition numbers 62368155 and 2370912."

I couldn't find .cif files for the refined structures on the CCDC neither with the provided submission codes, nor on the search by the authors or by the unit cell name. Please could the authors provide it as part of the submission package along with CheckCIF reports?

Could the authors also upload the raw diffraction frames to a filesharing service like Zenodo?

Version 1:

Reviewer comments:

Reviewer #1

(Remarks to the Author)

The authors have significantly improved the quality of the paper, responding to all the queries of the reviewers in my view.

The case for hexapoles is made more clear with the calculations on the electrostatics. I am still not convinced that the hexapolar description adds anything over an Ising type model, e.g. are there low energy excitations with multipolar character or would there be nearby phases with distinct hexapolar structures that would not arise for Ising spins? If the authors were to expand the Hamiltonian in orders of multipolar, from dipole through hexapole, it's not clear to me that the hexapolar term would be significantly different from zero. Nevertheless, I accept that the description, though not my preference, is valid so will leave it to the authors' judgement. I appreciate the authors highlighting of the interest in these multipolar behaviour: I hope that in future work they are able to demonstrate behaviour that is truly hexapolar.

Fig 4b and c is still confusing: why are the blocks split horizontally in some cases but not the others? Is this a rendering issue or am I missing some kind of distinction.

I appreciate the inclusion of additional data, and in particular, the pseudo-Bragg refinement suggested by another reviewer.

This paper should be accepted.

Reviewer #2

(Remarks to the Author)

The authors have done a good job in resolving not only my comments but also the rest of the criticisms received. I do not consider that additional changes are necessary.

Reviewer #3

(Remarks to the Author)

All my comments were answered, in my view the paper is ready for the publication.

Reviewer #1 praises the study and states that they 'think overall this paper will be a very valuable contribution to the literature'. We address the individual questions and comments in turn:

- The authors mention some of the other MOFs that feature disorder of an analogous cluster. They don't mention a number of other MOFs highlighted in Ref 54, principally those by the Eddaoudi group, or indeed disorder found in the NU family of 'shp' topology MOFs. This might help make the generality of the discovery clearer.

*We thank the reviewer for highlighting the articles from the Eddaoudi group which are, of course, highly pertinent to this study. We have included additional reference (61-63) and modified the text accordingly so that it now reads: "Previous examples of MOFs containing the RE9 SBU have been prepared using rigid linkers with little opportunity for alternative conformations. This class of MOFs, notably those with **shp** topology⁵⁴ have been widely studied and investigated for a variety of application. Despite this interest, disorder of the SBU has been commonly observed⁵⁴⁻⁶³ but no previous understanding of the correlation of the disorder has been proposed."*

- It took some effort to figure out the structure of the material from the main text: the ligand is never named and never shown clearly. I would suggest adding both. The composition of the cluster is also not specified clearly in the text. Further, the authors show that fluoride is likely in their MOF (as per Balkus <https://doi.org/10.1021/jacs.1c10493>), but there's no evidence of this in their reported composition.

We thank the reviewer for making this suggestion and have modified the figure accordingly – see Figure 1.

The reviewer is indeed correct that we show that there is fluoride present in the MOF. As discussed in the Supplementary Information the fluoride cannot be modelled from the crystallographic data, in part due the disorder of the system, and the clusters in particular, but also due to the similarity in electron density associated with either a fluoride or hydroxide. It is also likely that the fluoride is disordered over multiple sites in the cluster. Thus, we have not included fluoride in the composition of the MOF and note that it would make very little difference to the overall formula mass and other parameters. We thank the reviewer for suggesting the Balkus paper which we have included as a reference in the SI.

- What is the convention for J w.r.t. single and double counting of pairs. It is unclear from the text and SI.

This point has been clarified on page 6 of the revised manuscript.

- The authors rely on electrostatics and an 'hexapole' to justify the ordering along the c axis. This seems unlikely to be the real driving force, particularly as the authors show large changes in the molecular geometry of the ligand between orientations which lead to some energy differences. The authors also don't appear to relate the MC J with their DFT J. Using the synthesis temperature as 'T' gives I think $\sim 0.5 \text{ kJ mol}^{-1}$, which is surprisingly close to the estimate from DFT. Perhaps this sort of analysis could be included.

We thank the reviewer for raising this point for clarification and have added an additional paragraph to further elaborate and to include discussion of the electrotactic vs thermal contributions (see end of page 3). We have also added further discussion in the SI.

- What consequence does the hexapole have in the structure and modelling? It is not clear what additional insight this complexity brings. It also seems the authors model it as an Ising degree of freedom (and perhaps also refer to it as spin up and down in the SI?). This would probably be clearer to audiences less familiar with such high order multipoles.

We thank the reviewer for providing an opportunity to improve our description, and hence the reader's understanding, in this regard. In our opinion there are conceptual advantages of using hexapoles over spins:

1. *Conceptual: the hexapole depiction helps understanding of charge distribution and symmetry of the object. There is a parallel to f-orbitals, which will be informative to readers with a chemistry background.*
2. *Visual: we feel that the hexapole depiction helps the reader to understand why a rotation by 60 degrees is equivalent to "flipping" the spin.*
3. *Quantitative: electrostatics calculations, as suggested by the reviewer, rely upon the hexapole representation.*

We have addressed these points in revised text describing the SBU on page 3 of the manuscript whilst also addressing a further point from the reviewer – see below. We have also added further discussion in the SI.

- Could the authors estimate the energetic difference due to electrostatics and compare it to the DFT & MC derived energy?

See response to reviewer above and additional text added on page 3.

- Fig 2 would benefit from the cluster being shown on the same scale as the PDF.

We thank the reviewer for highlighting this point and have modified Figure 2 accordingly.

- "While the energy difference itself is not small compared to the thermal energy at the temperature of crystallization"
This discussion is a little confusing: perhaps it could be rephrased.

As both kT and J can be quantified we have modified the text so that it now reads: "While the energy difference itself is smaller when compared to the thermal energy at the temperature of crystallization ($J \sim 0.5 \text{ kJ mol}^{-1}$, $kT = 2.479 \text{ kJ mol}^{-1}$: $J < kT$),..."

- I don't understand Figure 3. Why are there 4 blocks in each stack when there are only 2 clusters in each stack. Figure 3 also implies the authors include the clusters in their DFT calculations, which they do not.

We thank the reviewer for highlighting the difficulties in appreciating (original) Figure 3. We have modified (what is now) Figure 4 to highlight the linker, in contrast to the SBUs, and the caption to include the statement "(note, DFT calculations were performed solely on the linker)".

- Minor issues: - "Although such structural disorder can be random, it is more often correlated.^{6,7}"
Correlated disorder is certainly common, but I don't think there is much evidence it is the majority of disordered systems. The references don't support it.

We thank the reviewer for raising this point and agree that the original statement was too strong. We have modified the text so that it now reads "Although such structural disorder can be random, it is often found to be correlated.^{6,7}"

- Reference 21 could be augmented by recent work from Chapman et al in JACS.

We thank the reviewer for suggesting the nice paper by Chapman et al and we have included the relevant citation (reference 22).

- "In principle, there are $C(18, 9) = 48620$ possible nodes with this correct number of Dy atoms" Is this right? It seems with the high site symmetry there are considerably less.

Thanks to the reviewer for raising this point which has helped to clarify our description. The site symmetry of the centre does not determine the symmetry of the disordered cluster. The single cluster configuration can lower the site symmetry, in the extreme case to 1.

We have reworded the text on page 3 of the manuscript as follows to clarify the point about possible configurations and also the choice of hexapole terminology as follows:

"The crystal structure of UoB-100(Dy) is highly disordered, as can be derived from the many partially occupied atom sites in the average structure refinement [Fig. 1(b, c)]. We interpret the crystal structure by first considering the metal-containing secondary building unit (SBU), which is centered around the 1b Wyckoff site (D_{6h} point symmetry). There are two symmetry-distinct Dy sites: 12o and 6m. Thus, there are 18 Dy atoms per SBU in total, although each of which has an occupancy of 0.5. In principle, when randomly selecting 9 out of 18 possible atom sites, there are $C(18,9) = \frac{18!}{9!(18-9)!} = 48620$ possible node configurations with the correct number of Dy atoms. However, on closer inspection, only two of these configurations are chemically feasible: the closest contact between the nearest 12o Dy sites is 2.21 Å and the closest contact between 12o and 6m Dy sites is 2.66 Å. We reason that it is unlikely that pairs of Dy atoms are separated by such short distances and therefore exclude node geometries that include these small separations. Only two configurations satisfy these local rules, as shown in Figure 1(d). We will go on to show experimentally that these nonanuclear clusters indeed exist in the local structure.

Besides the node disorder, it is clear from the average structure that the linker is disordered between two orientations: one where the linker points "up" along the c-axis, and one where it points "down". Since the scattering from the linker disorder is expected to be much weaker than the significantly heavier Dy atoms in the nodes, it is difficult to resolve correlations in linker orientation experimentally. For this reason, we solely focus on the dominant node disorder in our initial model.

The presence of the disordered nodes raises the question: how are these nodes distributed in space? To answer this question, we need to understand the interactions between the nodes. In our approach, we represent the local configurations of the nodes using hexapole charge distributions, preserving their D_{3h} point symmetry [Fig. 1(d)]. While a 60° rotation of these hexapoles is mathematically equivalent to the flipping of Ising spins, the hexapolar representations provide a clearer picture of the charge distribution of the nodes and, in turn, the electrostatic interactions between them (see S.I. for details). This disordered arrangement of multipoles draws comparison to magnetic multipole liquids $Ce_2Sn_2O_7$ and URu_2Si_2 ,⁶⁶⁻⁶⁹ as well as the assemblies of molecular multipoles in barocaloric plastic crystalline phases and molecular perovskites.^{70,71} In each case, understanding the correlated disorder present via diffuse scattering and modeling approaches, was crucial to understanding the systems."

We have also added further discussion in the SI.

- The authors say all relevant data is in the SI. I would strongly encourage them to be more expansive in the included data. In particular, the authors only include 2 cuts of their PDF data and total scattering in the main paper. Please can the authors include the DFT results as files, rather than just PDF tables as they are hard to parse.

We thank the reviewer for raising this point and agree with the suggestion. As noted below, in response to another reviewer we have now uploaded all the diffuse scattering data to Zenodo (see below) and have also added the DFT results as xyz coordinates as supplementary files.

- I would also encourage them to include the CIFs with the SI, as it saves a reader/reviewer going to the CSD to download them

We have submitted the cif files as part of the revised submission in addition to them being uploaded to the CSD (CCDC).

- What does spin up and spin down mean in the cluster geometry section of the SI?

This point is discussed in detail above.

- Typos: 0.20 kJ mol⁻¹

Corrected.

Reviewer #2 praises the study as a 'valuable contribution to the concept of correlated disorder in molecular architectures' stating that our study 'goes beyond complementing previous studies' and that 'this study has significant potential to attract attention and citations from colleagues working in the design and modification of MOFs and introduces a novel tool that will greatly benefit a smaller yet growing community interested in the structural analysis of defects in molecular solids'. We are very grateful to the reviewer for their kind words.

We address the individual questions and comments in turn:

- In the discussion on the linker, the authors describe its low symmetry as a design element to facilitate defect formation (by orientation?). I would recommend elaborating more clearly on the relationship between the connectivity/flexibility/symmetry of the linker and its effect on the generation of correlated defects.

We thank the reviewer for raising this point and have added a clarification on page 2 to aid understanding.

- Figure 1 does not clearly help to visualize the characteristics of the cluster and the linker. I would suggest adding a sub-figure to clearly illustrate this before introducing thermal factors or crystallographic disorder.

As noted above we have modified Figure 1.

- The text references the formation of up to 48,620 possible nodes with different positioning of Dy atoms in the Dy₉ cluster. How was this number calculated? Please clarify.

As noted above in response to reviewer #1 we have added further discussion about the description of the Dy₉ cluster.

- Prior to the discussion of the scattering data, there is an assumption regarding the possible distributions of Dy₉ nodes in space, driven by the presence of ferro- or antiferro-hexapolar charge distributions. In my opinion, it would be beneficial to delve deeper into the origin of these local interactions and whether these electrostatic interactions might extend to other clusters with

different symmetries commonly used in MOF design.

The reviewer highlights an intriguing aspect of our study which may feed into a wider understanding of correlated disorder. This is a potentially vast subject and therefore we have added a sentence to the article's conclusions suggesting that this may be a topic for future investigations. The added sentence reads 'The presence of hexapolar charge distributions in the Dy9 clusters also hints at a role for local electrostatic interactions in influencing local correlations. It will be beneficial to explore the potential role of such electrostatic interactions for other SBU configurations.'

- I do not consider this essential for publication; however, the study and its impact would greatly benefit from demonstrating how the distribution of antiferro- or ferro-hexapolar nanostructures might be modified with specific experimental parameters. Collecting scattering patterns at variable temperatures or in the presence of electric/magnetic fields could reveal a modulation in the distribution of nanodomains, which in other cases as UiO-type materials would only be accessible through synthetic variations.

The reviewer raises some intriguing questions and these will form the basis of future investigations.

Reviewer #3 praises the study and recommends publication. We address the reviewers individual questions and comments in turn:

- Could the authors add the enlarged diagram of the linker to the Figure 1? (same as on S1) Currently it is too small to understand it.

As noted above we have modified Figure 1.

- Figure 2a shows diffuse scattering with punched Bragg peaks from the main structure. Interestingly**, ** in some layers the punched Bragg peaks don't align with the plane they are expected to be (see attached image with the red line of hk6 plane). Are those peaks from a secondary phase?

We thank the reviewer for spotting this which we have amended in Figure 2. The cloudy diffuse features around punched Bragg peaks are likely the result from thermal contributions.

- Figure 2b shows a good fit of diffuse model to experiment. However, experimental scattering has peaks at low reciprocal space vector, while the model does not. Could the authors discuss potential source of this discrepancy?

This is a valuable point and we have added a sentence to reflect the discrepancy between the model and data at low-Q. As we note in the new text we did not include the linker disorder in these models and suggest that the differences may be attributable to linker ordering or relaxation.

- "Based on electrostatic arguments, it is expected that 'antiferro-hexapolar' interactions are favoured between such neighbouring node". From the experiment you have shown that in the ab plane the electrostatic argument is invalid, and it is the steric strain of the ligand which defines the sign of the nearest neighbour interaction. Could the authors discuss their expectations for preferred stacking along the c axis, based on the steric argument? Is there any obvious reason why ligands would prefer antiferro-ordering along c axis?

We thank the reviewer for giving us the opportunity to clarify this point. We have added additional text at the bottom of page 6 to clarify the role of electrostatic and steric effects.

- While it is commendable that the authors have done the diffuse scattering modelling, since diffuse scattering is well concentrated in half-layer peaks, did the authors try to integrate those diffuse *******peaks as Bragg peaks, and use those new peaks to refine the supercell of the structure? If successful, this can provide experimental insight into the relaxation of ligands.

We thank the reviewer for this suggestion and have performed the refinement of the supercell of the structure. The refinement was successful and has been included in the main text on page 5, including a new Figure (Figure 3).

- "these nodes assemble into a complex and novel nanodomain structure" - ****This appears to be**** an imprecise statement. First, the current paper doesn't give any direct access to the ***domain*** structure, only to local correlations which suggest that such domain structure should probably exist with anisotropic shape of the domains. I suggest removing this sentence.

Thanks to the reviewer's suggestion to perform a supercell refinement we have been able to provide more insight into the domain structure.

- Typos: Page 1: Cu_x Mn (should it be Cu_x Mn_{1-x} ?)

Interestingly both forms are used in the relevant papers but we have changed the formula as suggested.

- in SI some references are missing, they are just indicated as (REF)

We thank the reviewer for spotting this omission and have added the references.

- Data availability: " The X-ray crystallographic coordinates for structures reported in this study have been deposited at the Cambridge Crystallographic Data Centre (CCDC), under deposition numbers 62368155 and 2370912."

I couldn't find .cif files for the refined structures on the CCDC neither with the provided submission codes, nor on the search by the authors or by the unit cell name. Please could the authors provide it as part of the submission package along with CheckCIF reports?

We are sorry about this. The cifs were present on the CCDC but it seems that they had to be found through the Referee Service function. The cifs and checkcifs have been submitted with the revised version.

Could the authors also upload the raw diffraction frames to a filesharing service like Zenodo?

Thank you for this suggestion. We intended to do this but weren't sure what the best method or location was for sharing the data. We have now uploaded the data to Zenodo - DOI: 10.5281/zenodo.14269933.

Response to reviewer's comments

Reviewer #1 (Remarks to the Author):

The authors have significantly improved the quality of the paper, responding to all the queries of the reviewers in my view.

The case for hexapoles is made more clear with the calculations on the electrostatics. I am still not convinced that the hexapolar description adds anything over an Ising type model, e.g. are there low energy excitations with multipolar character or would there be nearby phases with distinct hexapolar structures that would not arise for Ising spins? If the authors were to expand the Hamiltonian in orders of multipolar, from dipole through hexapole, it's not clear to me that the hexapolar term would be significantly different from zero. Nevertheless, I accept that the description, though not my preference, is valid so will leave it to the authors' judgement. I appreciate the authors highlighting of the interest in these multipolar behaviour: I hope that in future work they are able to demonstrate behaviour that is truly hexapolar.

Fig 4b and c is still confusing: why are the blocks split horizontally in some cases but not the others? Is this a rendering issue or am I missing some kind of distinction.

We have added to the Figure 4 caption to clarify this question such that the caption now reads as follows, with the new text highlighted in yellow:

'Figure 4 The nanostructure of UoB-100(Dy). (a) Representation of one of the nanostructures of UoB-100(Dy), as produced by the MC procedure detailed in the text. The colour of the hexagons represents the SBU orientations as shown in Figure 1(d). Note that there is order along *c*, and nanoscale order in the *ab* plane. **Black/white blocks represent single SBUs in different orientations, as shown in Figure 1, which are ordered in our model along the *c*-direction.** (b) The structure which represents an antiferrohexapolar arrangement of the SBUs in the plane and DFT-relaxed linker arrangement constrained by these adjacent node geometries. (c) The same arrangement for ferrohexapolar neighbours. The constrained linker geometry optimizes with a lower energy, indicating that this ferrohexapolar node arrangement leads to less strain. (note, DFT calculations were performed solely on the linker). (b, c: atom colours, C, black; N, blue; O, red; H, pink).'

I appreciate the inclusion of additional data, and in particular, the pseudo-Bragg refinement suggested by another reviewer.

This paper should be accepted.

Reviewer #2 (Remarks to the Author):

The authors have done a good job in resolving not only my comments but also the rest of the criticisms received. I do not consider that additional changes are necessary.

Reviewer #3 (Remarks to the Author):

All my comments were answered, in my view the paper is ready for the publication.

a)

$h0l$

8

Expt.

Expected Braggs plane

4

What is this?

l